# Analysis of the Properties of Hardox Extreme Steel and Possibilities of Its Applications in Machinery

Beata Białobrzeska [1,*] , Robert Jasiński [1] , Łukasz Konat [1] and Łukasz Szczepański [2]

1 Department of Vehicle Engineering, Faculty of Mechanical Engineering, Wrocław University of Science and Technology, Wybrzeże Wyspiańskiego 27, 50-370 Wrocław, Poland; robert.jasinski@pwr.edu.pl (R.J.); lukasz.konat@Pwr.edu.pl (Ł.K.)
2 Faculty of Mechanical Engineering, Wrocław University of Science and Technology, Wybrzeże Wyspiańskiego 27, 50-370 Wrocław, Poland; lukasz.szczepanski@pwr.edu.pl
* Correspondence: beata.bialobrzeska@pwr.edu.pl; Tel.: +48-71-320-3845

**Abstract:** The article presents the results of Hardox Extreme steel tests in the as-delivered state from a steel mill (after quenching and tempering), and also in the normalized state. The research procedures included a microstructure analysis using light microscopy; and a static tensile test at ambient temperature to determine its Young's modulus, yield strength, tensile strength, elongation and reduction in area after fracture. During the tensile tests, both the longitudinal and transverse orientation of rolling direction were taken into account. The Charpy impact test was also carried out in the temperature range of the ductile–brittle transition in connection with the fractographic analysis carried out with the use of a scanning microscope (SEM). The impact tests were carried out on samples in both directions on the plate, using the following temperatures: $-40$, $-20$, $0$, $+20\,^\circ C$. Based on the structural and strength characteristics of Hardox Extreme steel determined on the basis of the research, in a further part of the paper the possibility of its use in machine construction elements operating in selected industrial sectors is considered/discussed, with a particular emphasis on reducing the level of energy consumption in the manufacturing and operation of the above technical facilities.

**Keywords:** martensitic boron steel; heat treatment; mechanical properties; structures; fractographic analysis; Hardox Extreme steel; machinery



## 1. Introduction

Industrial development in such economy sectors as power engineering, mining, civil engineering, agriculture and transport generates increased greenhouse gas emissions, thereby resulting in disadvantageous climate changes. Heavy goods vehicles alone account for 27% of $CO_2$ emissions in road transport and nearly 5% of greenhouse gas emissions in the whole European Union [1]. Particular emphasis on limiting global $CO_2$ emissions in practice influences all economy sectors. For example, in 2019 the European Parliament adopted regulations reducing the allowed $CO_2$ emission quota for high capacity vehicles by 15% by 2025 and by 30% by 2030. From 2025 manufacturers will have to meet the target of at least 2% market share of zero-emission and low-emission vehicles in the total new vehicles sales. Moreover, according to the Paris Agreement on climate change, before 2022 the European Commission has to propose a quota for the period following the year 2030 [2]. One of the methods used to limit greenhouse gas emission in heavy load carrying vehicles and machinery may be, for example, replacing mineral fuels with biofuels and the development of electric motors, whose use is not possible yet in such heavy equipment, however. Another idea is the reduction of power consumption in construction itself by applying modern, high-strength materials, allowing one to decrease the cross-sections of selected elements in vehicles and machinery. One of the most common materials is still steel which is 100% recyclable. It is estimated that the world's demand for steel will grow

from 1800 million tonnes in 2018 to 2500 million tonnes in 2050 [3]. Currently, annual global $CO_2$ emissions related to the steel industry total about 2.8 billion tonnes, which accounts for 7% of the total $CO_2$ emission. The pressure on limiting greenhouse gas emissions refers also to metallurgy. For the purposes of meeting the targets specified in the Paris Agreement, metallurgy-related $CO_2$ emissions should be reduced to the level of 400–600 million tonnes per year in 2050 [4]. World leaders in blast furnace capacity, with simultaneous low $CO_2$ emissions, are the Swedish and Finnish steel industry sectors. Swedish steel producer SSAB AB provides modern, abrasion resistant Hardox steel, which belongs to boron steels with increased abrasion resistance, or a group referred to as martensitic boron steels. However, other manufacturers meet customer expectations by producing modern boron steels with increased abrasion resistance, such as: ThyssenKrupp Steel Europe AG (XAR and TBL steels), Dillinger Hütte GTS (Dillidur steels), Grobblech GmbH (Durostat), AcelorMittal (Usibor), TATA Steel Group (Abrazo), TITUS Steel (Endura), SUMITOMO Metal (Sumihard) and JFE EVERHARD Corporation (JFE-EH).

The history of research on boron as an alloy addition is very long. Its beginnings date back to 1907; however, at the time the amount of boron added to steel was too large (0.2–1.5%) and it resulted in the intensive liberation of boron compounds, which led to a decrease in mechanical properties [5]. At the next stage of research, very small amounts of boron were used by Walter in 1921. This allowed him to obtain self-quenching steels with satisfactory mechanical properties [6]. However, these steels were not used to their full potential due the lack of research results repeatability caused by insufficient metallurgic purity of steel. It was not until 1935 that a ferro-alloy, added to steel as a deoxidiser by an American steel mill, accidentally contained elements which prevented boron from binding with oxygen and nitrogen. As a result, a group of weldable bainitic steels with a tensile strength of 530–1200 MPa was obtained. In the years that followed, the element was used increasingly more often to replace expensive alloy additions, such as nickel, molybdenum and chromium, when wars significantly restricted the availability of these elements [7]. In Europe, low-alloy boron steels with increased abrasive wear resistance were produced for the first time in 1970 exactly by the Swedish steel company SAAB—Oxelösund. Hardox 400, produced then for the first time, is a steel characterized by low carbon content, simple chemical composition, non-complex microstructure and high hardenability, achieved owing to the addition of boron. Currently, the Hardox steel family encompasses nine steel types. These are modern materials with good strength and technology-related properties; additionally, they can be manufactured using energy-efficient production lines. Interestingly, in 2016 SSAB started the HYBRIT project (Hydrogen Breakthrough Ironmaking Technology), in which the main energy source is fossil-free electricity. In 2017, the initial feasibility study was done; it confirmed that the proposed development path was technologically feasible and economically justified, and in February 2018 the decision to build pilot installation was made [4]. All these characteristics mean that the Hardox steels can be used in numerous economic sectors, e.g., to make modern, high load-carrying and simultaneously low mass constructions, which results in decreasing manufacturing costs by limiting the use of input materials and operating costs. The very SSAB steel mill, in its advertising material, places considerable emphasis its activity targeted at $CO_2$ emission reduction, called EcoUpgraded. Some of the examples of efficient actions are the use of, among others, Hardox steel in haul truck linings, allowing one to reduce fuel consumption by 13,500 L throughout the time of use, which means saving 49 tonnes of $CO_2$ [8]; and the use of abrasion resistant steel in a tipper vehicle lining, thereby saving 67,500 L of fuel, which translates to 220 tonnes $CO_2$ [9].

New Hardox steel types are characterized by higher hardness. Hence, it is possible to formulate the thesis that the hardness index constitutes the main parameter quoted by the manufacturer to forecast and increase abrasion resistance in Hardox steel. These actions have some influence also on plastic material properties, which in the case of Hardox steel applications must be comprehensively analyzed together with abrasion resistance. In the present study, the main goal was the determination of basic structural and strength

indexes for the Hardox Extreme steel, with consideration of the longitudinal and transverse orientation in the thermo-plastic modification of a plate and the most frequently used temperature range in impact tests, from −40 to +20 °C, allowing one—in the majority of engineering applications—to determine the ductile to brittle transition curve. It is worth emphasizing that Hardox Extreme steel is characterized by a tensile strength of 2000 MPa, declared by the manufacturer, which is obtained with carbon content of less than 0.50% by weight, which assumes major significance in its use to combine welding techniques. The above statement is supported by the advertising-type information supplied by the manufacturer indicating that Hardox Extreme is the hardest wear plate in the world, which, when specialist tools are used, is susceptible to cutting, milling and drilling; another strongly emphasized property is that it can be used in welding processes [10]. However, in the above-mentioned information on the discussed steel, there are no details related to its properties. This is an essential factor justifying further research on this type of steel. Taking into consideration the possibility of using welding techniques to join the analyzed steel, which means structural and strength changes, a set of tests for Hardox Extreme steel was performed for both the delivery state from a steel-mill and after normalizing. According to the authors, the use of this technological operation and the selection of appropriate conditions and parameters is critical for the successful performance of Hardox Extreme steel welding processes. That is why in the present study—regardless of the state in which the analyzed steel was delivered from a steel mill—the evaluation was done at the same level for its structural strength properties both in the delivery state (after quenching) and in the normalized state. However, due to their very complex nature, the issues related to the welding and heat treatment of the Hardox Extreme steel are discussed in a separate study underway.

## 2. Materials and Methods

In the structural and tensile tests, a Hardox Extreme steel plate in the delivery state directly from a steel mill was used. The dimensions of the heat-treated steel plates were approximately 300 mm × 220 mm × 10 mm. The samples assigned for particular research procedures were cut out of the plate with a margin of at least 20 mm using a high energy abrasive water jet and electroerosion. All thermal processes took place in gas-tight chamber furnaces FCF 12SHM/R (Czylok, Jastrzębie-Zdrój, Poland) in an ambient gas atmosphere—99.95% pure argon. The normalization processes encompassed: austenitizing at 800 °C for 60 min and air cooling. The heat treatment parameters were selected based on experimental data and CCT and TTT graphs (Figure 1), which were drawn up for the purpose of the study. They were obtained via computer simulation using JMatPro software (Sente Software Ltd, Surrey, UK). For the purpose of maintaining an optimum cooling rate, thermal processing was conducted on larger plate fragments, and at a later stage appropriate samples for research were taken from them, maintaining the margin related to higher cooling intensity on the edges. During normalization, the surface temperature of the sheets was controlled by a pyrometer. With respect to required dimensions and roughness tolerances, at the final stage all the samples were smoothed.

The chemical composition analysis was done by the spectral method using an emission analyzer with a glow-discharge GDS500A (Leco Corporation, St. Joseph, MI, USA) The following parameters were used during the analysis to allow for ambient ionization: U = 1250 V, I = 45 mA, 99.999% argon. The obtained results were the arithmetic averages of at least five measurements.

Microstructural observations were made using a light microscope, Nikon Eclipse MA200 (Nikon Corporation, Tokyo, Japan) coupled with a digital camera, Nikon DS-Fi2 (Nikon Corporation, Tokyo, Japan). The tests were conducted on samples etched in a 3% solution of $HNO_3$ as per the ASTM E407 standard. The recorded images were analyzed and saved using NIS Elements software (Nikon Corporation, Tokyo, Japan).

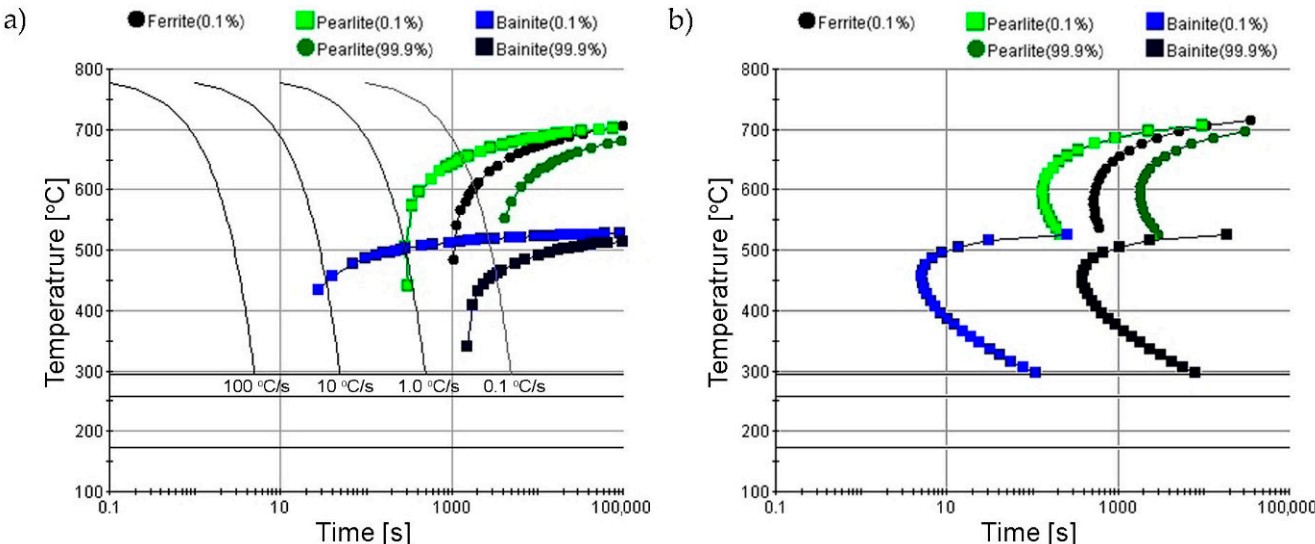

**Figure 1.** Time–temperature graph for steel with chemical composition (% by mass): C—0.44, Si—0.16, Mn—0.49, P—0.006, S—0.002, Al—0.04, Cr—0.83, Cu—0.02, Co—0.02, Ni—2.01, Mo—0.14, B—0.0021, Ti—0.008, V—0.008; austenitized at temperature $T_A$ = 787 °C; assumed size of the former austenite grain—10 μm. (**a**) Constant cooling—CCT. (**b**) Isothermal cooling—TTT. Assigned temperatures for individual transformations, phases and components of the structure: pearlite −720 °C, ferrite −736 °C, bainite −531 °C, martensite (50%) −258 °C, martensite (90%) −173 °C, $M_S$ = 294 °C.

The hardness measurements of the tested samples were done with a universal hardness tester Zwick/Roel ZHU 187.5 (Zwick Roell Gruppe, Ulm, Germany). The using the Rockwell method, as per the PN-EN ISO 6508-1:2016-10 standard. Five hardness measurements were performed on each of the samples whose microstructure had been previously assessed.

The tensile tests were conducted in ambient temperature on rectangular, proportional fivefold samples, as per PN-EN ISO 6892-1:2020-05. A material testing system MTS 810 (Mts Systems Corporation, MN, USA) was used with an extensometer, measurement base length $L_0$ = 25 mm. During the tests, the stretching rate was controlled by the stress increase rate. Next, the following were determined for each sample: Young's modulus ($E$), elastic limit ($R_{p0.05}$), yield strength ($R_{p0.2}$), tensile strength ($R_m$), the percentage reduction of area ($Z$) and percentage elongation ($A_5$) after tear. The strength indices are the arithmetic averages of the results obtained from at least five samples per each measurement point. Additionally, on the basis of the results obtained for particular samples, measurement errors were calculated in the form of standard deviation.

The impact tests were done using a hammer, Charpy Zwick Roell RPK300 (Zwick Roell Gruppe, Ulm, Germany). The initial energy value was 300 J, as per PN-EN ISO 148-1:2017-02. The samples used in the tests were square with V-shaped notches. On the basis of the research done at +20, 0, −20 and −40 °C temperatures, the arithmetic average of at least five samples and the standard deviation for the impact tests were determined for each measurement point. Subsequently, fractographic analyses were conducted on selected fracture surfaces using a scanning microscope (SEM) JSM-6610A (JEOL Ltd., Tokyo, Japan). The tests were done at an accelerating voltage of 20 kV. Observations were made in the material contrast, using SE detectors (JEOL Ltd., Tokyo, Japan). On the basis of the recorded images of crack surfaces, the percentage share of plastic zones was calculated using free image analysis free software ImageJ version 1.52a (National Institute of Mental Health, Bethesda, MD, USA).

### 3. Results

#### 3.1. Chemical Composition Analysis

The chemical composition of Hardox steel greatly depends on plate thickness; the reason why is that the goal is obtaining a high hardness level with similar values over the whole cross-section area, which simultaneously is a derivative of the uniform material structure. The real chemical composition of Hardox Extreme for a 10 mm thick plate is shown Table 1. The analyzed material belongs to the medium carbon steel group (0.44% C). Attention should be drawn to the contents of such elements as: chromium (0.83%), nickel (2.01%) and molybdenum (0.14%) which intensify the influence of boron, whose content is 0.0021%. Simultaneously, it should be noted that boron content is close to its limit value (0.0025%), thanks to which in low alloy (low carbon) steels, an increase in hardenability is observed [11–14]. Boron, when added in larger amounts, binds with $Fe_2B$, whose particles become embryos facilitating the diffusion transformation, as a result of which hardenability decreases. It should be added that the efficiency of boron in steel depends also on the form in which this element occurs and is strictly related to the chemical composition and metallurgic purity of steel; this especially refers to carbon and nitrogen contents [15]. This is the reason why the analyzed steel was enriched with such additions as titanium and aluminum—elements making more lasting compounds than nitrides and boron. The limit of carbon content whereby a positive influence from boron on hardenability can be observed is 0.53% [12]; however, alloy additions intensifying boron's impact may decrease the carbon content level to the point at which a hardenability increase is observed. An element which prevents it is manganese, occurring in the analyzed steel at 0.49%.

**Table 1.** Contents of chemical elements in Hardox Extreme steel (wt.%).

| C | Mn | Si | P | S | Cr | Ni | Mo |
|------|------|------|-------|-------|------|------|------|
| 0.44 | 0.49 | 0.16 | 0.006 | 0.002 | 0.83 | 2.01 | 0.14 |
| **V** | **Cu** | **Al** | **Ti** | **Nb** | **Co** | **B** | **Zr** |
| 0.008 | 0.018 | 0.043 | 0.008 | 0.001 | 0.024 | 0.0021 | - |

Another element, apart from boron, which improves hardenability is chromium; it is frequently used with a toughening element such as nickel to produce superior mechanical properties. Nickel does not form any carbide compounds in steel; it remains in solution in the ferrite, thereby strengthening and toughening the ferrite phase [16].

An important alloy addition in the Hardox Extreme steel is also molybdenum, whose task is to prevent temper brittleness, which may be enhanced by the joint occurrence of such elements as chromium and nickel. Due to the fact that Hardox Extreme steel is designed to be joined using welding techniques, the occurrence of irreversible temper brittleness is possible in heat affected zones. Moreover, it was found out that Hardox Extreme steel contains very minor harmful additions, mainly of sulfur and phosphorus, which may confer the possibility of very high strength indices while maintaining plastic properties.

#### 3.2. Microstructure Analysis

The information on the structure of Hardox Extreme steel in the delivery state provided by the manufacturer is not very precise. In the catalogue card [10], in the State of Delivery section there is information that the steel is delivered in the hardened state. However, in the Production and Other Recommendations section, more details are added—namely, that Hardox Extreme steel reaches its mechanical properties by quenching, and if necessary, by yet another tempering process. The tempering process in high-strength steels has a decisive influence on their plastic properties, particularly impact properties, and also the presence and performance of diversified failure mechanisms which can be identified during fractographic analyses.

The microstructures of Hardox Extreme steel in the delivery state and after normalization are shown in Figures 2–5. The structure of steel in the delivery state is medium-carbon

tempered martensite with visible boundaries of prior austenite grains (Figure 3a,b). A characteristic feature of the martensite in low and medium-carbon steels is its three-level hierarchy in morphology—it is composed of laths, blocks and packets. Martensite laths making a block are characterized by the same crystallographic orientation, and hence they represent the same variant of the created martensite structure. Moreover, in both directions of rolling direction, one can observe bright bands, which indicates slightly increased carbon content in these areas. The bands are longer and wider in the case of longitudinal direction (comp. Figure 2a,b).

After normalization, in both the longitudinal and transverse directions relative to the rolling direction, the whole spectrum of structures was obtained, in which quenching sorbite and quenching martensite prevailed (Figure 5a,b). In numerous areas, the sorbite structure indicates a post-martensitic structure—tempering sorbite. Martensitic regions are banded and run to the directions of the thermomechanical rolling and are related to the bright bands of the martensitic structure observed in the delivered condition.

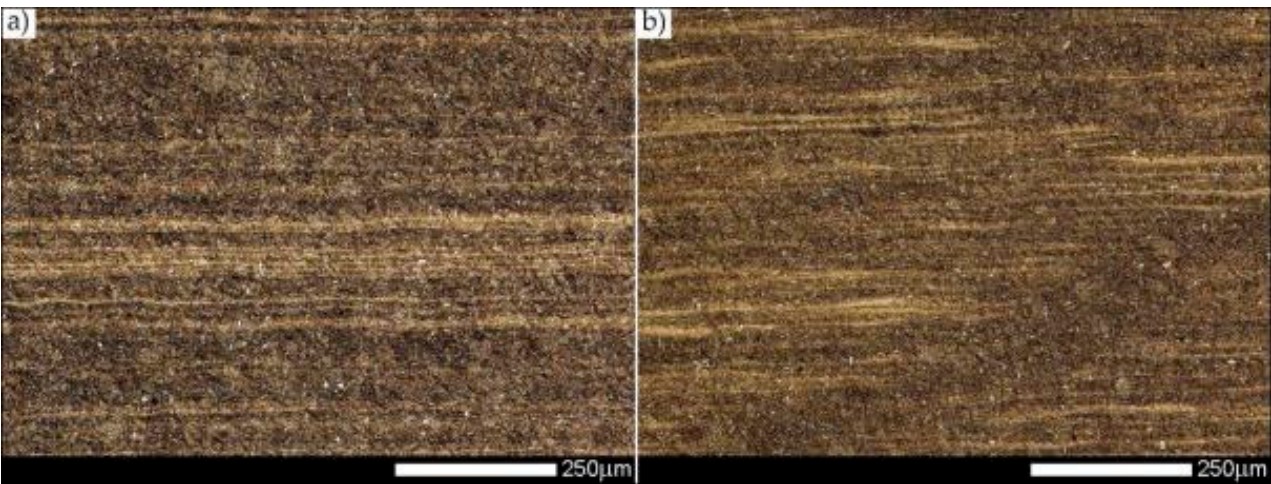

**Figure 2.** Hardox Extreme steel microstructure in the delivery state with clearly distinct banding pattern resulting from thermomechanical rolling. (**a**) Longitudinal orientation—hardness 60 HRC. (**b**) Transverse orientation—hardness 61 HRC. Light microscopy, etched with 3% $HNO_3$.

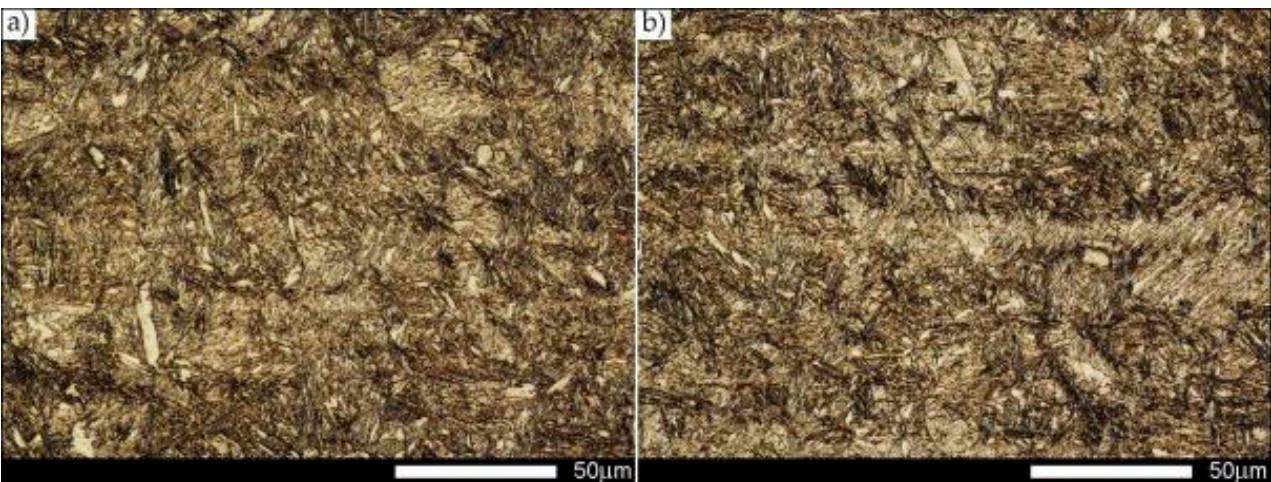

**Figure 3.** Magnified image of microphotography central zone shown in Figure 2. (**a**) Longitudinal orientation; (**b**) transverse orientation. Quenched and tempered martensite—two-dimensionally—with fine-lath morphology structure. Light microscopy, etched with 3% $HNO_3$.

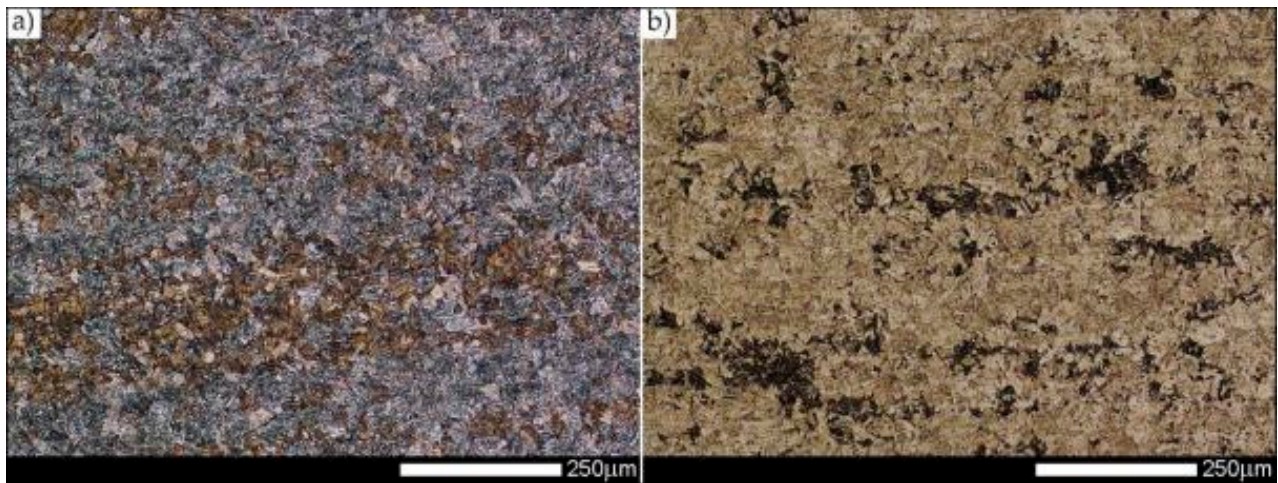

**Figure 4.** Hardox Extreme steel microstructure in normalized conditions. (**a**) Longitudinal orientation—hardness of 38 HRC. (**b**) Transverse orientation—hardness of 38 HRC. In both cases, quenched structures exhibit wide structural diversity with a great percentage of diffusion structures. Light microscopy, etched with 3% $HNO_3$.

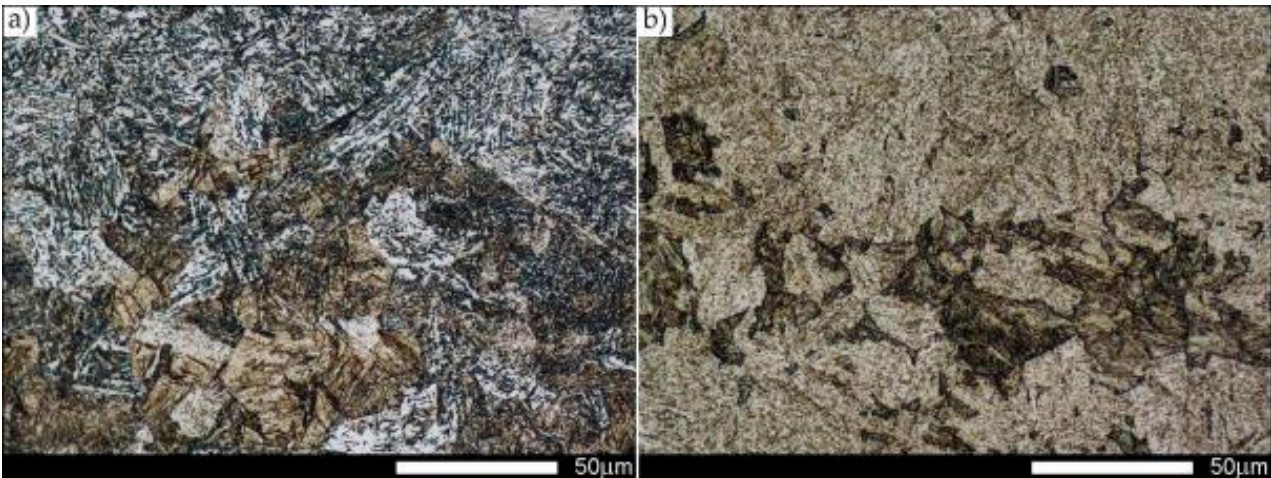

**Figure 5.** Magnified images of the central zones of the microphotography shown in Figure 4. (**a**) Longitudinal orientation. (**b**) Transverse orientation. The microstructures consisting of quenching sorbite and quenching martensite. Light microscopy, etched with 3% $HNO_3$.

### 3.3. Static Tensile Test

The tensile tests' results for Hardox Extreme steel samples are presented in Table 2 and Figure 6. They allow one to conclude that in the delivery state—for both directions of plate forming process—the examined steel is characterized by very high static tensile strength indices ($R_m$), significantly above 2000 MPa. However, it should be noted that in the case of the transverse orientation the value of $R_m$ was approximately 300 MPa lower. The analyzed steel was characterized by a relatively low value of yield strength $R_{p0.2}$; unlike strength, it did not exceed 1600 MPa, and contrary to strength, its value was higher for the transverse orientation of the plate. From a utilitarian point of view, the indices determining the ratio of the yield strength to the tensile strength ($R_{p0.2}/R_m$) seem interesting. The value of the ratio, i.e., 0.64–0.74, confirms the susceptibility of this steel to significant strengthening after exceeding the yield strength and does not entail the necessity to use high safety factor values in constructions, which will definitely influence the reduction in their energy consumption. The structural analysis and the chemical composition analysis (the small number and weight fraction of alloy elements with high affinity for carbon)

showed that the high strength values resulted from two main strengthening mechanisms. They are performed during the forming process. Thermomechanical rolling is one of them, which, when combined with a large number of alloy micro-additions (mainly increasing recrystallization temperature), leads to the fragmentation of austenite grains and the high carbon saturation of ferrite, which results from the high hardenability achieved owing to boron micro-addition.

**Table 2.** Selected mechanical properties of Hardox Extreme steel: D—state of delivery, N—normalized state, L—longitudinal orientation, T—transverse orientation.

| Sample | $E$ | $R_{p0.05}$ | $R_{p0.2}$ | $R_m$ | $A_5$ | $Z$ | $R_{p0.2}/R_m$ | HRC |
|---|---|---|---|---|---|---|---|---|
| | GPa | MPa | MPa | MPa | % | % | - | |
| D(L) | $210 \pm 8$ | $1150 \pm 60$ | $1549 \pm 28$ | $2413 \pm 116$ | $3.5 \pm 0.9$ | $10.1 \pm 2.2$ | 0.64 | $60 \pm 0.1$ |
| D(T) | $206 \pm 6$ | $1252 \pm 33$ | $1574 \pm 22$ | $2117 \pm 85$ | $1.5 \pm 0.9$ | $8.0 \pm 1.1$ | 0.74 | $61 \pm 0.1$ |
| N(L) | $203 \pm 12$ | $660 \pm 51$ | $871 \pm 86$ | $1256 \pm 132$ | $10.5 \pm 0.7$ | $33.1 \pm 3.5$ | 0.69 | $38 \pm 0.6$ |
| N(T) | $212 \pm 1$ | $687 \pm 161$ | $863 \pm 118$ | $1130 \pm 16$ | $11.3 \pm 0.8$ | $36.8 \pm 2.7$ | 0.76 | $38 \pm 0.2$ |

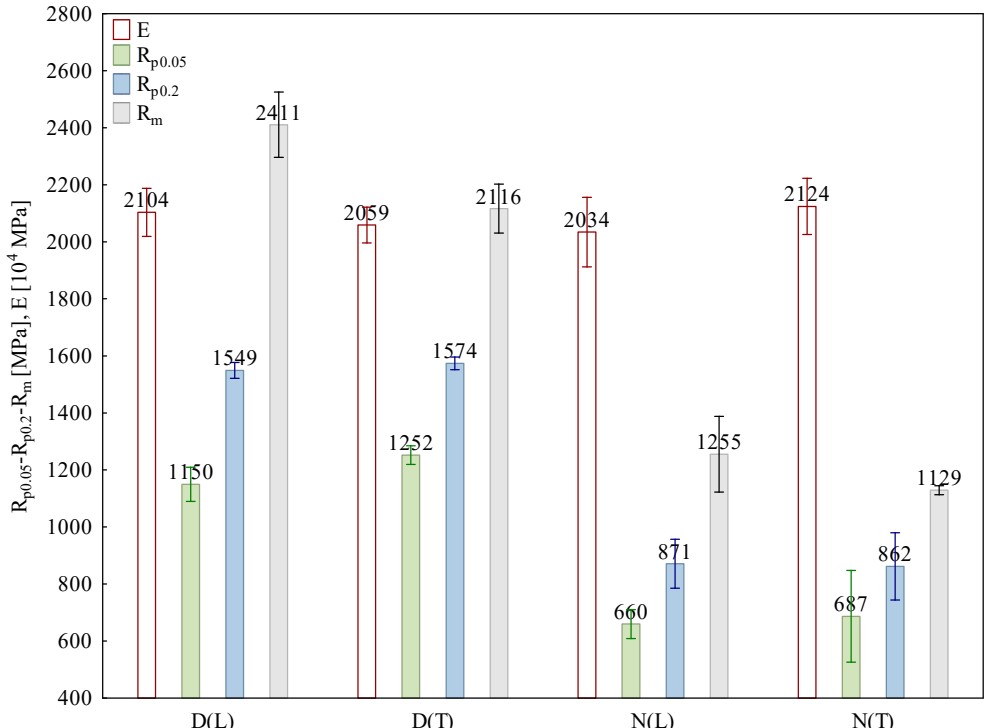

**Figure 6.** Tensile strength ($R_m$), yield strength ($R_{p0.2}$), elastic limit ($R_{p0.05}$) and Young's modulus ($E$) of Hardox Extreme steel in delivery and normalized states.

Very good strength properties of Hardox Extreme steel in the delivery state unfortunately correspond with unsatisfactory plastic properties, represented by the reduction in area ($Z$) and the elongation ($A_5$). The values of apparent elongation were 3.5% for the longitudinal orientation and 1.5% for the transverse orientation; in the case of contraction they were 10.5% and 11.3%, respectively (Figure 7).

Normalization led to significant reductions in strength properties. Taking into account the orientation of forming process, the value of strength decreased by 47–48%, the yield strength by 44–45% and the elastic limit by 43–45%. Simultaneously, it resulted in increases in the parameters determining plastic properties, i.e., elongation and reduction in area by 200–653% and 228–360%, respectively (Figure 7).

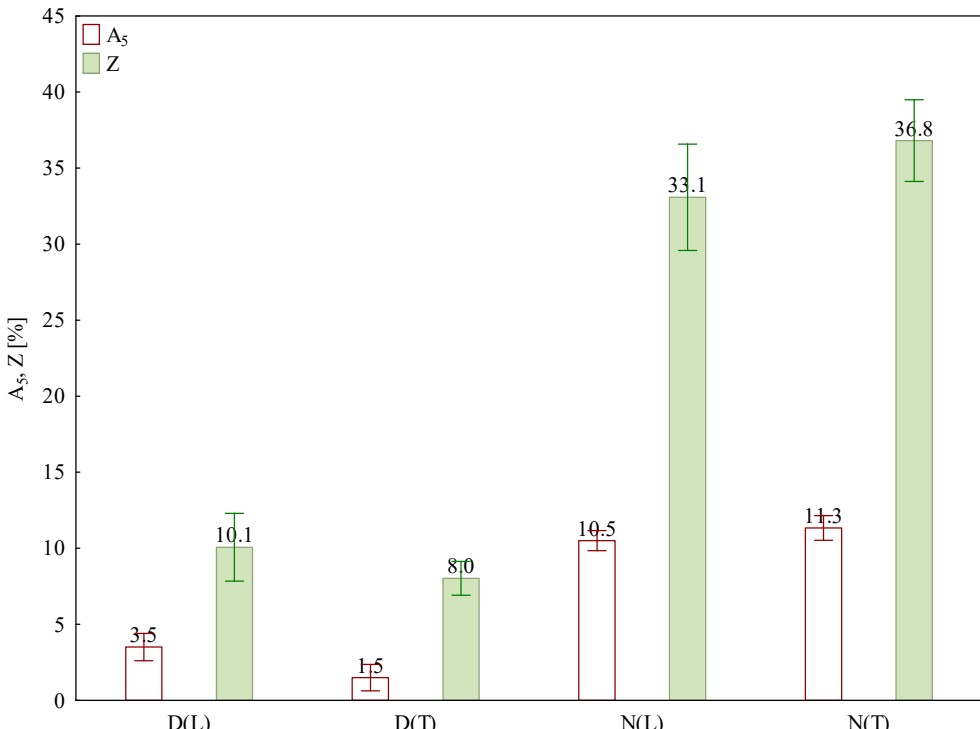

**Figure 7.** Elongation ($A_5$) and reduction in area ($Z$) at break of Hardox Extreme steel in delivery and normalized states.

*3.4. Impact Tests*

The impact tests results are presented in Figure 8. In the case of construction steels the most frequently adopted brittleness point is the quantity criterion, corresponding with the value of break work of 27 J or impact strength of 35 J/cm$^2$, which correlates with its occurrence in the middle of the brittle and ductile fracture [17]. The analyzed steel neither meets the above criterion in any heat treatment state, nor meets it at any test temperature. However, it is worth mentioning that in the recorded course of impact strength changes in Hardox Extreme steel in the delivery state, there was no significant reduction in impact toughness. In the case of the delivery state, the differences in the impact strength between the forming process orientations are not large. Hence, despite the use of thermomechanical rolling, the materials show a certain isotropy of properties. After the impact strength test conducted in ambient temperature, the impact strength in the forming process transverse orientation was only 7.3% lower than in the longitudinal orientation. The most significant difference of over 20%, was recorded for the impact tests conducted at a temperature of −20 °C; however, it should be added that the samples taken in the transverse orientation were characterized by higher impact strength. In the case of the longitudinal orientation, the relative difference between the highest and the lowest impact strength value was about 50%; the lowest value was obtained for the test temperature of −20 °C. By analogy, for the transverse orientation, the relative impact strength difference in the whole test temperature range was just over 40%; however, in this case the lowest impact strength value was obtained for the lowest test temperature. In the normalized state, the impact strength values are higher than in the delivery state, which in the case of Hardox steels was recorded earlier for Hardox 400 and Hardox 500 [18]. The highest impact strength of 14.6 J/cm$^2$ was obtained for the normalized state, in the longitudinal orientation, after the impact strength test conducted in ambient temperature. Similarly to the case of the delivery state, the impact strength differences depending on forming process orientation are insignificant. After the test conducted at a temperature of −20 °C, the impact strength values after normalization were the same for both the longitudinal and transverse orientations.

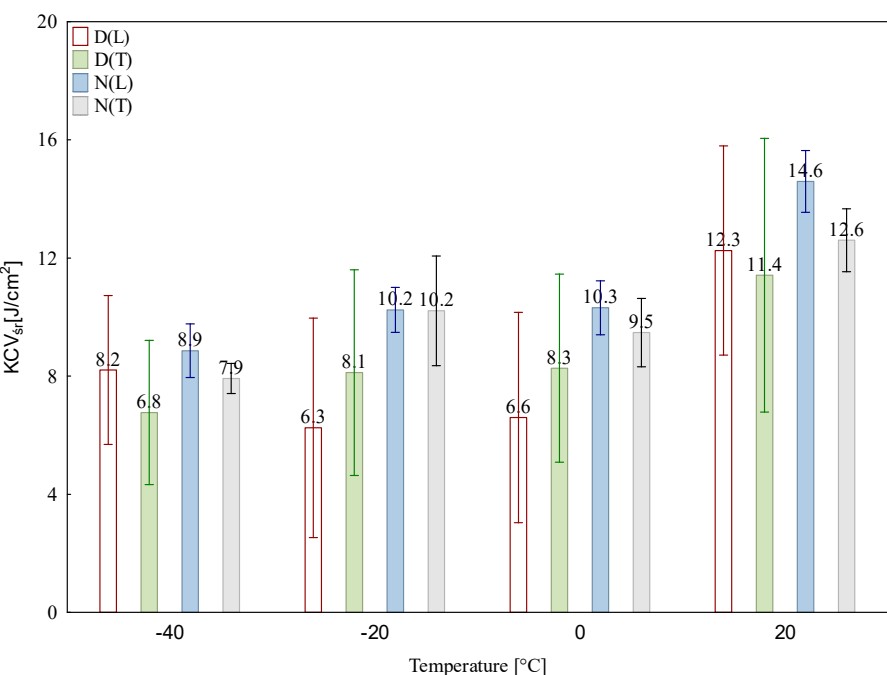

**Figure 8.** Changes in the impact strength of Hardox Extreme steel as a function of temperature in delivery and normalized states.

### 3.5. Fractographic Analysis

In impact tests, it is extremely essential to analyze the nature of a fracture, because the adopted criterion of 35 J/cm$^2$ (discussed above) corresponds with the occurrence of a ductile and brittle fracture. Figures 9 and 10 present the macroscopic images of fractures in the analyzed steel after the impact strength test at temperatures of +20 and −40 °C. The areas which were analyzed in detail later in the research are marked (Figures 11–18). In each case, the fractures in the delivered and normalized states exhibited the lack of plastic deformation and practically no participation of ductile side zones and the zone below the notch. In the delivered state, the participation of ductile zones was from about 5% (longitudinal orientation, impact strength test at −40 °C) to about 8% (longitudinal orientation, impact tests at +20 °C). After normalizing, the plastic zone was practically not observed. This situation resulted in the reduced impact strength of this steel in the whole range of test temperatures.

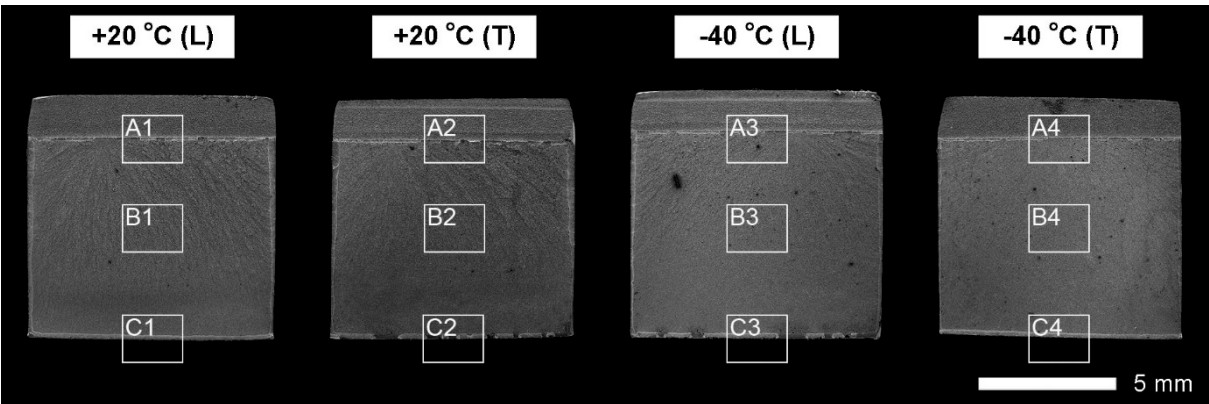

**Figure 9.** Macroscopic image of fractures in representative specimens of Hardox Extreme steel in delivery state: L—longitudinal orientation, T—transverse orientation; frames are used to mark: A—zone below the notch, B—central zone, C—final-fracture zone. SEM non-etched state.

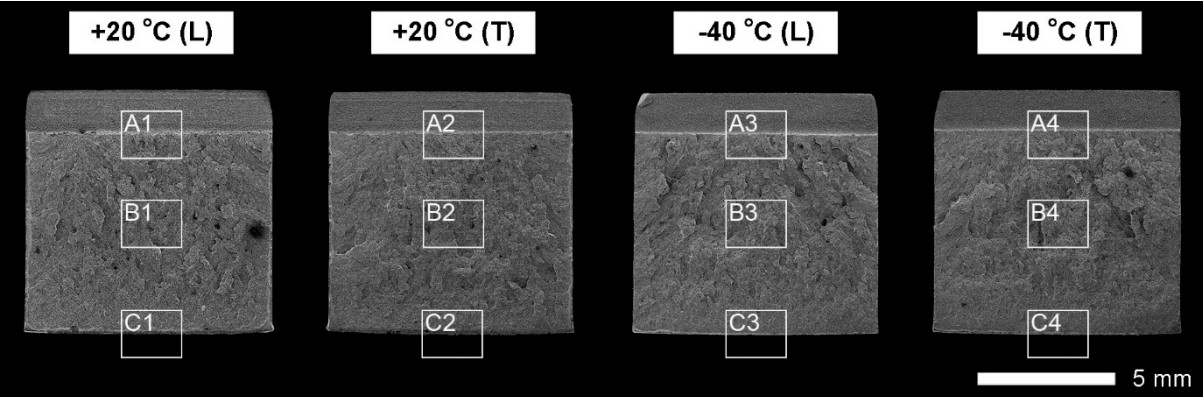

**Figure 10.** Macroscopic image of fractures in representative specimens of Hardox Extreme steel in a normalized state: L—longitudinal orientation, T—transverse orientation; frames are used to mark: A—zone below the notch, B—central zone, C—final-fracture zone. SEM, non-etched state.

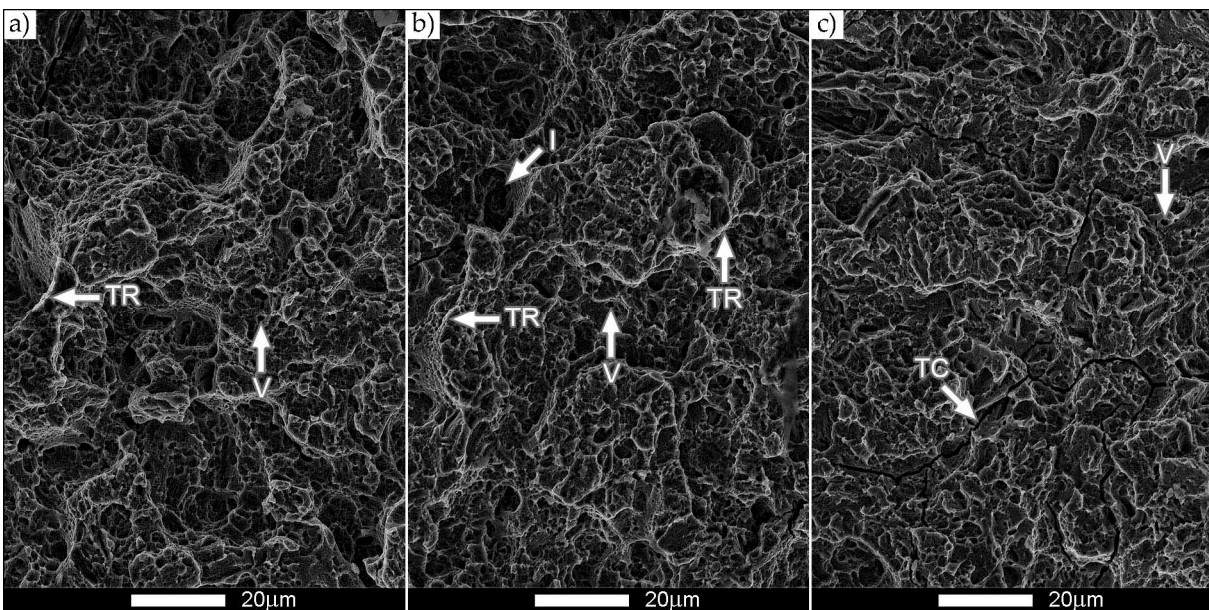

**Figure 11.** Images of the Hardox Extreme steel fracture surface in delivery state shown in Figure 9: longitudinal orientation (+20 °C). (**a**) The area marked with the A1 frame. (**b**) The area marked with the B1 frame. (**c**) The area marked with the C1 frame. TR—tear ridges, V—microvoids, I—inclusions, voids left after debonded inclusions, TC—transversal cracks. SEM, non-etched state.

The microscopic analysis (SEM) of fractures revealed the details and characteristic features in their structure. In the delivery state, for all samples (Figures 11–14), even those which underwent the impact strength test at the lowest temperature, the fracture classified as a quasi-tear was obtained. Its characteristic features are extensive topography, transverse cracks—visible especially in Figures 11c, 12c and 13a–Figure 14b—and voids left after the carbide phases are separated from the matrix (Figures 11–13). Apart from the places left by small carbide phases, there were also large voids probably left by non-metallic inclusions (Figures 11b and 13b). The quasi-brittle fracture occurs as a result of brittle cracking in small, local areas, followed by combining them together into one crack surface due to plastic deformation. Facets are small and partly plastically deformed. Although the facets are similar to brittle facets, owing to the occurrence of "river" sculpting, the identification of crystallographic surfaces, along which the cracking process takes place, is practically impossible. The meandering "river" system creates voids, whose structure resembles a ductile fracture, on a large surface [17,19]. The ridges of quasi-brittle facets, which create

a system of elevations and voids, are typical ductile surfaces. A change of the character of this fracture can be described exactly on basis of the participation of and ductile ridges. Their participation in the fracture surface decreased with reducing the test temperature (Figures 12 and 14). The quasi-brittle fracture is a typical fracture, observed earlier in martensitic steels [20,21]. Thus, the low impact strength in the delivered state was the effect of the lack of plastic zones below the mechanical notch and in sample side zones.

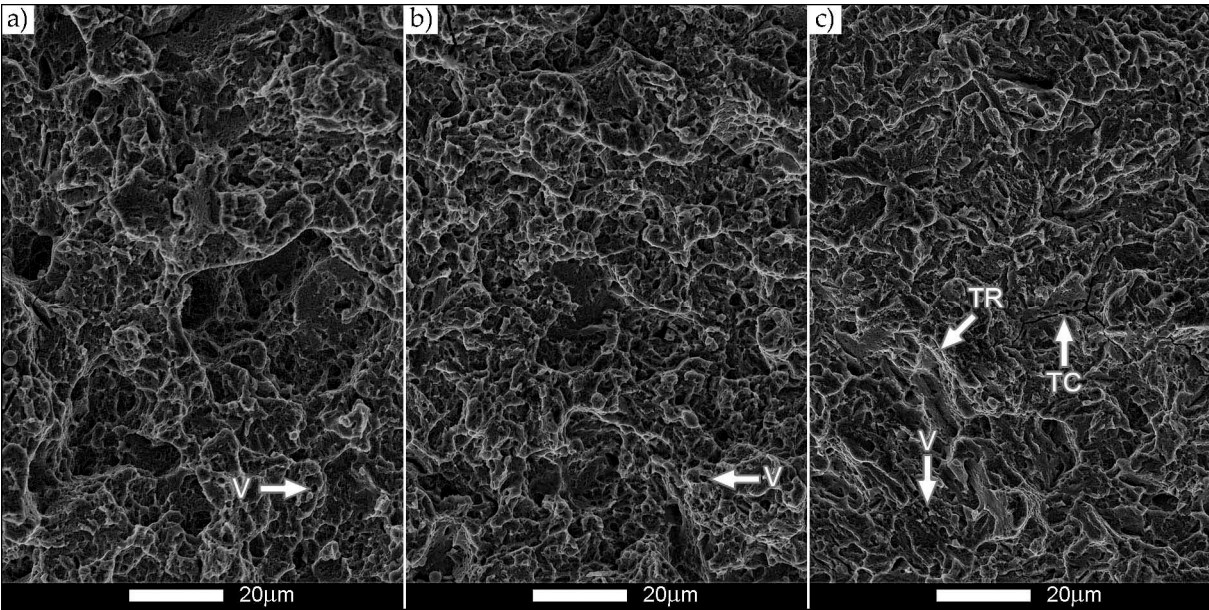

**Figure 12.** Images of the Hardox Extreme steel fracture surface in delivery state shown in Figure 9: longitudinal orientation (−40 °C). (**a**) The area marked with the A1 frame. (**b**) The area marked with the B1 frame. (**c**) The area marked with the C1 frame. TR—tear ridges, V—microvoids, TC—transversal cracks. SEM, non-etched state.

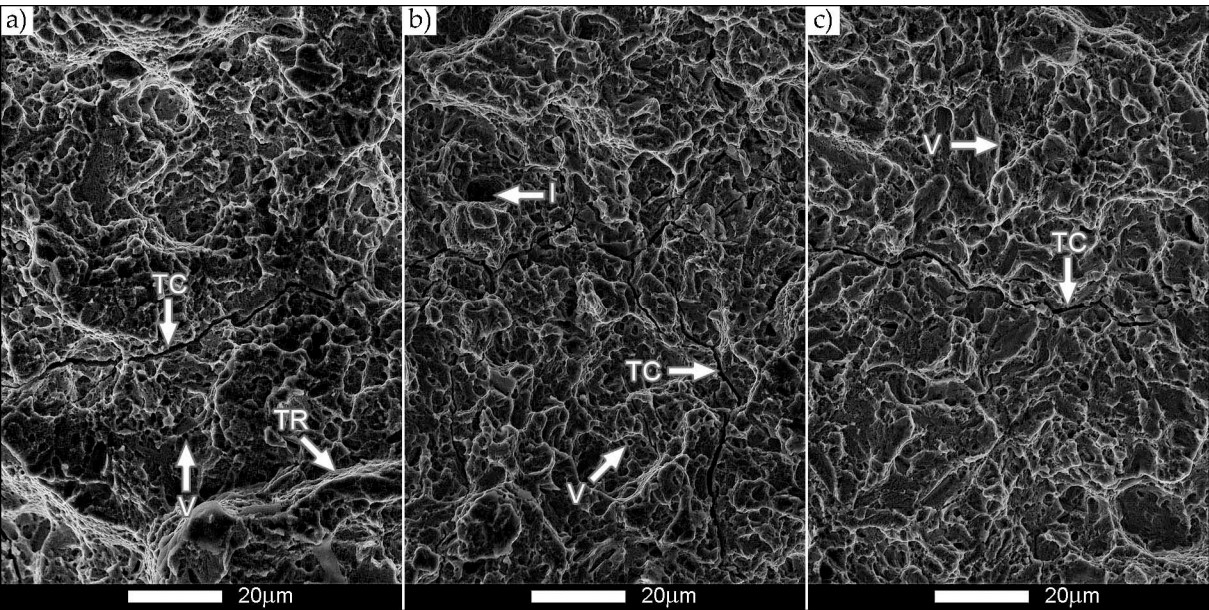

**Figure 13.** Images of the Hardox Extreme steel fracture surface in delivery state shown in Figure 9: transverse orientation (+20 °C). (**a**) The area marked with the A1 frame. (**b**) The area marked with the B1 frame. TR—tear ridges, V—microvoids, I—inclusions, voids left after debonded inclusions, TC—transversal cracks. (**c**) The area marked with the C1 frame. SEM, non-etched state.

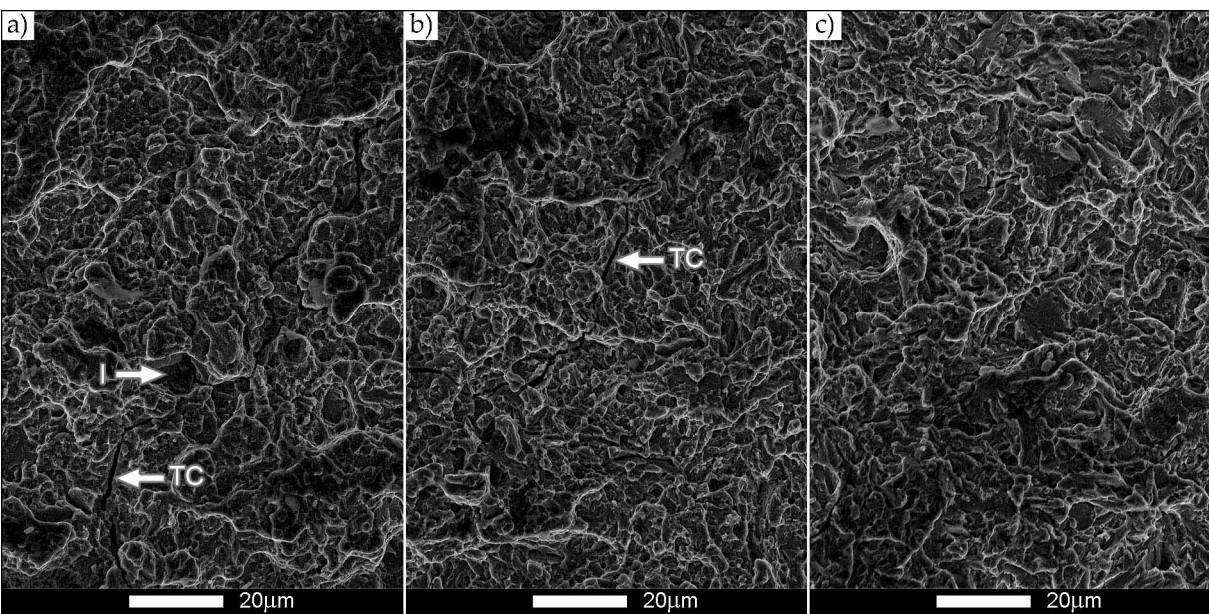

**Figure 14.** Images of the Hardox Extreme steel fracture surface in delivery state shown in Figure 9: transverse orientation (−40 °C). (**a**) The area marked with the A1 frame. (**b**) The area marked with the B1 frame. (**c**) The area marked with the C1 frame. I—inclusions, voids left after debonded inclusions, TC—transversal cracks. SEM, non-etched state.

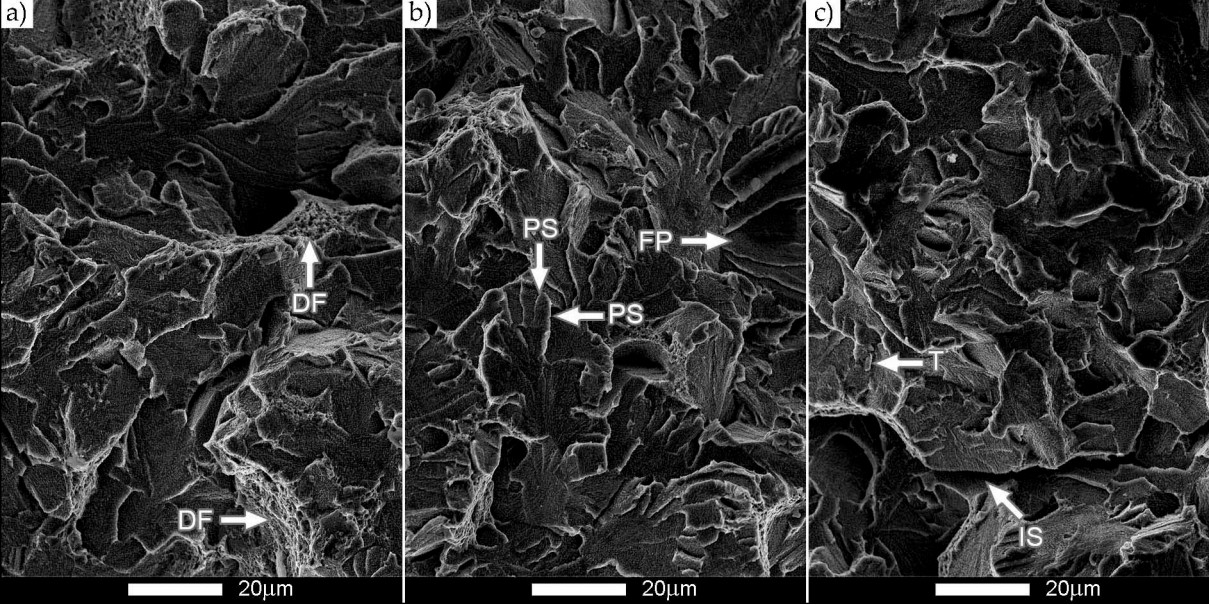

**Figure 15.** Images of the Hardox Extreme steel fracture surface in normalized state shown in Figure 10: longitudinal orientation (+20 °C). (**a**) The area marked with the A1 frame. (**b**) The area marked with the B1 frame. (**c**) The area marked with the C1 frame. DF—ductile fracture, PS—parallel steps, FP—fan pattern, T—tongues, IS—intergranular steps. SEM, non-etched state.

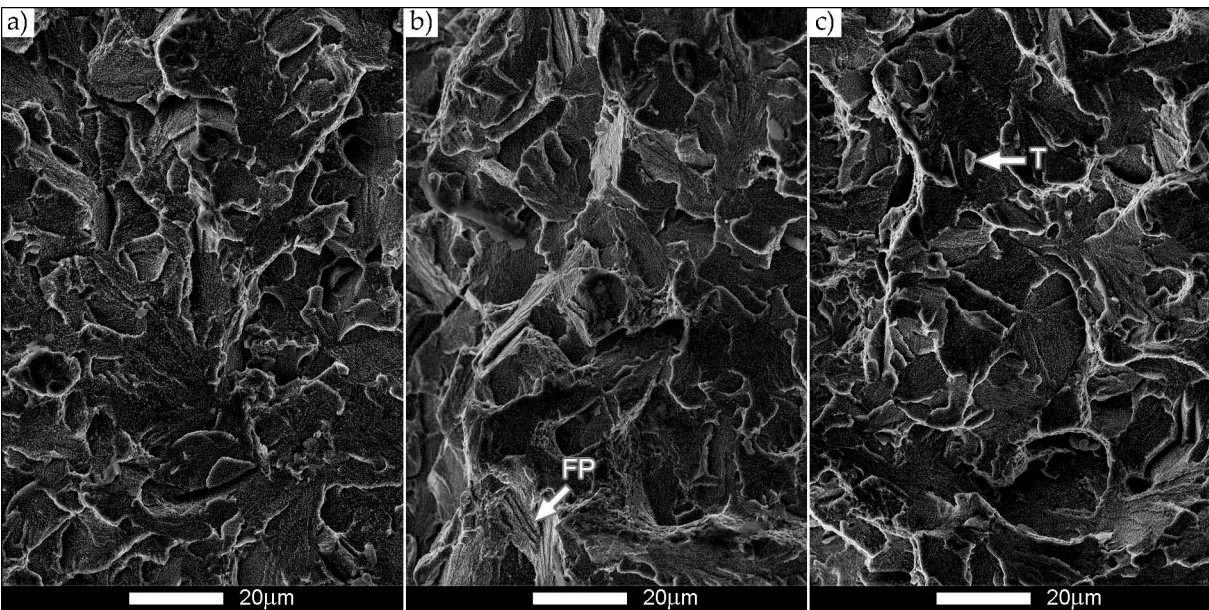

**Figure 16.** Images of the Hardox Extreme steel fracture surface in normalized state shown in Figure 10: longitudinal orientation (−40 °C). (**a**) The area marked with the A1 frame. (**b**) The area marked with the B1 frame. (**c**) The area marked with the C1 frame. FP—fan pattern, T—tongues. SEM, non-etched state.

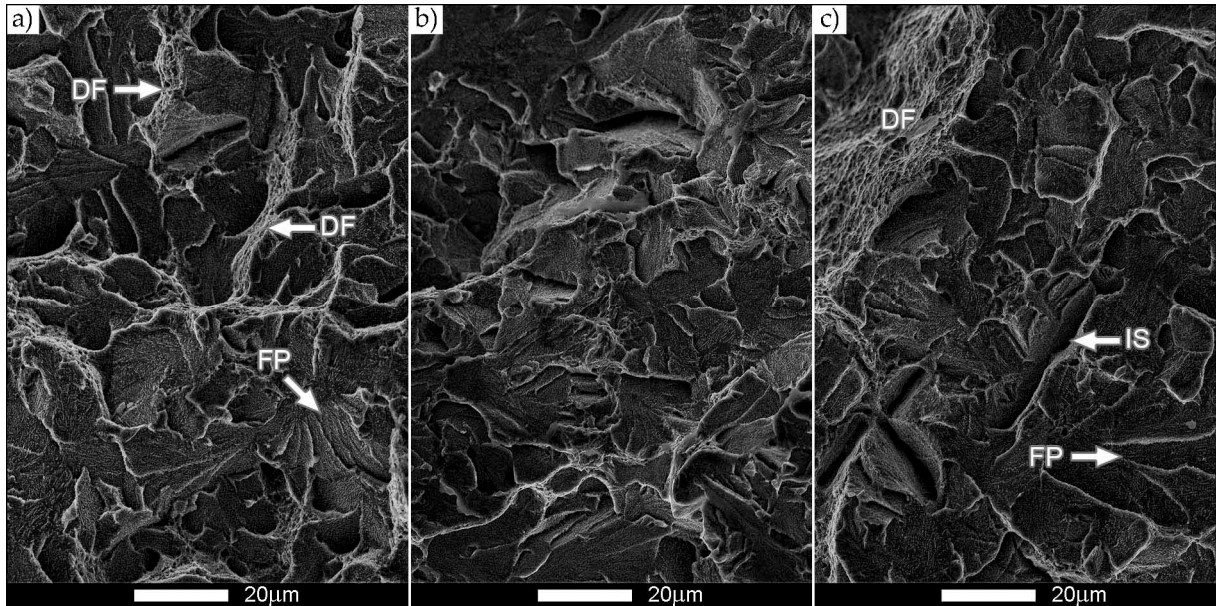

**Figure 17.** Images of the Hardox Extreme steel fracture surface in normalized state shown in Figure 10: transverse orientation (+20 °C). (**a**) The area marked with the A1 frame. (**b**) The area marked with the B1 frame. (**c**) The area marked with the C1 frame. DF—ductile fracture, FP—fan pattern, IS—intergranular steps. SEM, non-etched state.

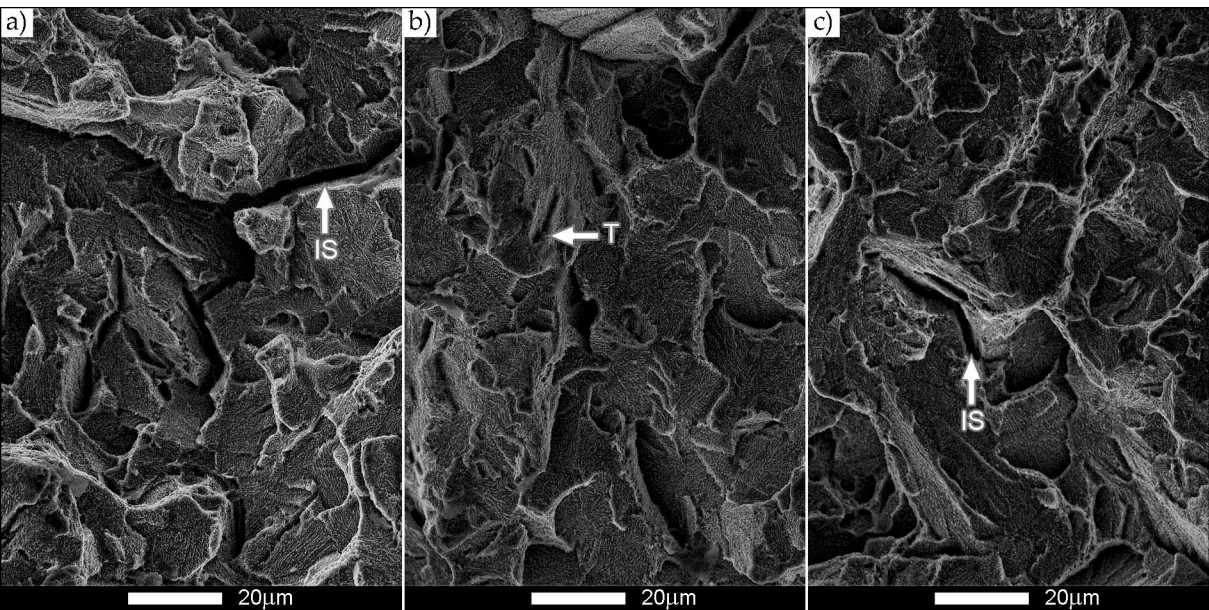

**Figure 18.** Images of the Hardox Extreme steel fracture surface in normalized state shown in Figure 10: transverse orientation (+20 °C). (**a**) The area marked with the A1 frame. (**b**) The area marked with the B1 frame. (**c**) The area marked with the C1 frame. IS—intergranular steps, T—tongues. SEM, non-etched state.

After normalizing, the fracture structure in the Hardox Extreme steel samples changed in comparison with the delivery state. Although the normalized samples exhibited higher impact strength, fracture surfaces had the structure typical of a transcrystalline brittle fracture (Figures 15–18). On the surfaces of fractures, there were brittle facets and characteristic river sculpted steps, with transcrystalline crack traces. Large steps occurred on grain boundaries (Figures 15c, 17c and 18a,c), and smaller ones occurred within grains (Figures 15b, 16b and 17a,c). Their presence means that the crack did not take place in one crystallographic plane; it bounced from one plane to another. According to [19], this effect occurs when the fracture front encounters a screw dislocation, and the step is size is conditioned by the size of the Burgers vector. The presence of steps also influences the change of the direction of crack propagation. As a result, in some sections crack growth is delayed, and this merges neighboring steps and connects them into a "river" system or a "fan" pattern (Figures 15b, 16b and 17a,c). In some places, the steps of brittle surfaces are arranged in parallel (Figure 15b). Another characteristic feature is the short elevations, described in the literature as "tongues" (Figures 15c, 16c and 18b). Following impact tests conducted in ambient temperature, aside from typical brittle facets, small ductile fracture areas occurred (Figures 15a and 17a,c). They result from the fact that in polycrystalline bodies, there are always a small number of grains with orientation or stress states which are unfavorable for brittle fracture occurrence [22,23].

Despite the fact that the Hardox Extreme steel obtained lower impact strength values in the delivery state than in the normalized state, the presence of fine-lath morphology in its structure initiated the creation of a quasi-brittle fracture, made of small, partly plastically deformed facets. A fracture propagating in this way ensured a higher safety level before the occurrence of a brittle fracture in comparison with the state after normalizing.

### 3.6. Analysis of Possible Hardox Extreme Steel Applications

A rational approach to the construction and material solutions in selected machinery elements, when based on real research results, may significantly extend useful life or even expand the exploitation possibilities of the whole operating facility. For example, the selection of materials used to make scoops in basic lignite mining equipment is one of the basic factors limiting its mining capacity. Due to hard operating conditions, these elements have to be prematurely replaced, which entails high costs, resulting from both material

and time aspects. However, the durability of particular elements depends not only on the types of materials used, but also on heat treatment, joining technology and various mining conditions. A good example here is the excavator bucket teeth used in opencast mining, in easily mineable overlays—sandy clay: due to their high hardness, martensitic structures show the highest wear resistance. In the case of land which is hard to mine, the dominance of dynamic loads should be indicated, and simultaneously, one must acknowledge the prevalence of the sorbite structure preventing cutting blades from breaking. However, such an approach does not ensure sufficient durability of a work blade. The above example shows that with regard to insufficient impact (dynamic) properties, certain caution is needed in the Hardox Extreme steel applications. The key to the determination of the Hardox Extreme steel's use areas, despite its low impact strength, is the $R_{p0.2}/R_m$ ratio at the level of 0.64–0.74, which is similar to that of other construction steels. This yield strength to tensile strength ratio allows one to use the Hardox Extreme steel in solutions with low safety factors. It should be noted here, however, that a high $R_{p0.2}/R_m$ ratio is most frequently indicated by the constructors of basic lignite mining equipment as the main index—apart from weldability—limiting the use of high-strength, low alloy steels in the construction of such equipment.

Another issue which influences the limited range of applications of the Hardox Extreme steel (similarly to other steels with analogous properties from various manufacturers) is related to its weldability. The results of the basic research and fractographic analysis reported in this paper show that the Hardox Extreme steel in the normalized state—from the qualitative perspective—exhibited a significant degradation of its performance properties in comparison with the delivery state. In consequence, the direct use of the steel in this heat treatment state is not recommended, and is even irrational. Nonetheless, this more extensive research undertaken by the authors with regard to the use Hardox Extreme steel in selected welding technologies showed very favorable influences (although in the context of weldability—critical) of the structures obtained as a result of normalizing on the process of welding and shaping the mechanical and useful properties of the welded joints of this steel owing to additional heat treatment processes. As mentioned above, the discussed issues will be discussed in detail in a separate study.

When considering the possible Hardox Extreme steel application areas on the basis of the conducted research, one should take into account application solutions in which the used of the investigated material may not be justified. This mainly refers to the most heavily loaded cutting elements—made of high alloy tool steels or sintered carbides in the powder metallurgy process, which from the technological perspective currently are very intensively developing and there is no direct alternative for them among weldable steels. However, despite certain limitations, the conducted research allows one to indicate a broad range of potential application of this steel, which would create opportunities for reducing machine exploitation costs resulting from both direct wear of work items and technological downtime. Simultaneously, thanks to the possibility of using the extremely high strength indices of the Hardox Extreme steel (decreased active cross sections of construction elements), it is worth considering also lower energy consumption costs, which also limit the emission of harmful chemical compounds. The types of Hardox steel currently used in vehicles and machinery encompassing, e.g., Hardox HiTuf, Hardox 400, Hardox 450 and Hardox 500, according to the information on the manufacturer's website, allow one to make significant fuel savings, and hence limit $CO_2$ emissions. Applications recommended in Hardox Extreme's information and marketing material are various types of lining constructions and sheets, e.g., in minerals mining [10]. Taking into account the obtained strength and impact tests results, on the condition that welding issues are resolved, apart from applications recommended by the manufacturer, the Hardox Extreme steel application areas may also encompass:

1.  In the mining industry:

    - Feeder system linings;
    - Linings for vibration baskets and basket drawers;

- Excavator bucket cladding and sheathing;
- Highly loaded areas of fixed buckets wheel chute with a block crusher.

2. In the transport industry:

- Walls and floors in semitrailers, trailers and containers;
- Floors and boards of tipper kippers and haul trucks;
- Concrete mixer drums.

3. In construction industry:

- Cladding and sheathing for excavator buckets, loaders, crushers, and scissors;
- In agriculture;
- Ploughshares;
- Blades;
- Coulters.

In the above list one element, namely, the teeth (and partly also blades) of basic lignite mining equipment buckets were intentionally unmentioned; the reason why is that they are the element most significantly limiting the durability of these constructions. In this case, only based on construction steels, one should not expect a material solution which could radically increase their durability. This results mainly from complexity in terms of material effort operating conditions. All of the above mentioned proposed Hardox Extreme steel applications undoubtedly require operation tests. The obtained laboratory research results definitely encourage starting such tests. However, it is worth noting that such research has already been largely conducted for the Hardox 400 and Hardox 500 steels. The experimental results presented in this work [24] encompassed the use of liner sheets made of the above-mentioned steels, instead of the hitherto used P355N steel, with an abrasion resistant coating in the fixed chute of a lignite excavator KWK-1500s (operating in the Turów coalmine in Poland). After over 1400 h of machine operation, the results turned out to be promising. This particularly refers to Hardox 500. Simultaneously, during the experiment, the welding problems previously signaled in this work with reference to the Hardox Extreme steel, were also observed, especially the durability aspect. Therefore, the defined problems related to the use of welding techniques to join the Hardox Extreme steel, and also other types of steel belonging to the same group, have been confronted in practice in actual implementations in heavy engineering machines and in numerous laboratory research results. For this reason, undertaking scientific considerations, signaled in the study on the Hardox Extreme steel related issues, seems by all means justified. In addition to this, the authors take the view that there is no possibility of conducting extended research on the discussed steel, for example undertaking the fracture mechanics, without determining its real structural and strength properties, established during basic research and fractographic analysis. On the margin of these considerations on the use of high strength, weldable, low alloy steels in machine constructions elements, it is worth noting that there is a necessity to change the attitude to designing selected structural nodes, or whole constructions. Such actions have been observed for over ten years in the context of using Hardox steels as structural steels by SSAB, in real—monocoque—constructions of transport vehicles used to carry mineral resources. Such an approach was possible only based on the use of very high strength indices of steel (and taking them into account in in the design process), which characterize martensitic boron steels resistant to abrasive wear.

## 4. Conclusions

1. The structure of the Hardox Extreme steel in the delivery state constitutes medium carbon, fine-lath martensite with hardness of 60 HRC (longitudinal orientation) and 61 HRC (transverse orientation). In the normalized state, its structure with a hardness of 38 HRC (longitudinal and transverse orientation) is mainly composed of quenched sorbite and martensite.

2.  The analyzed material belongs to medium carbon steel group (0.44%C). Attention is directed to the alloy content of such elements as: chromium (0.83%), nickel (2.01%), molybdenum (0.14%) and boron (0.0021%).
3.  In the delivery state the Hardox Extreme steel is characterized by the following mechanical parameters: $R_m$—2411 MPa, $R_{p0.2}$—1549 MPa, $R_{p0.05}$—1150 MPa, $E$—210 GPa, $A_5$—3.5%, $Z$—10.1% (longitudinal direction); $R_m$—2116 MPa, $R_{p0.2}$—1574 MPa, $R_{p0.05}$—1252 MPa, $E$—206 GPa, $A_5$—1.5%, $Z$—8.0% (transverse direction).
4.  In the normalized state, the Hardox Extreme steel has the following mechanical indices: $R_m$—1255 MPa, $R_{p0.2}$—871 MPa, $R_{p0.05}$—660 MPa, $E$—203 GPa, $A_5$—10.5%, $Z$—33.1% (longitudinal direction); $R_m$—1129 MPa, $R_{p0.2}$—862 MPa, $R_{p0.05}$—687 MPa, $E$—212 GPa, $A_5$—11.3%, $Z$—36.8% (transverse direction).
5.  Fractures in the delivery state and after normalization show the lack of macroscopic plastic deformation in each case, and practically no participation, or very small participation, of ductile side zones and the area below the notch.
6.  The fractographic analysis showed that after normalization, the fracture exhibits the structure typical of transcrystalline brittle fracture, made of brittle facets, while in the delivery state the obtained fracture is classified as quasi-brittle in which facets are plastically deformed.
7.  The extremely high strength indices obtained in the research, despite the not completely satisfactory plastic indices, offer the Hardox Extreme steel extensive application opportunities. The use of this type of steel will allow one to significantly reduce fuel consumption and $CO_2$ emissions due to the reduction of vehicle mass, which, especially in the transport industry, will allow one to comply with the regulations introducing $CO_2$ emission quotas for high capacity, heavy goods vehicles.

**Author Contributions:** Conceptualization, B.B., Ł.K. and R.J.; methodology, Ł.K., R.J. and Ł.S.; validation, Ł.K. and R.J.; formal analysis, B.B., Ł.K. and R.J.; investigation, B.B., Ł.K., R.J. and Ł.S.; resources, Ł.K.; data curation, Ł.K., R.J. and Ł.S.; writing—original draft preparation, B.B. and Ł.K.; writing—review and editing, B.B. and Ł.K.; visualization, Ł.K. and B.B.; supervision, Ł.K. and R.J.; project administration, Ł.K.; funding acquisition, Ł.K. All authors have read and agreed to the published version of the manuscript.

**Funding:** This research received no external funding.

**Data Availability Statement:** Data is contained within the article.

**Acknowledgments:** The authors would like to thank Józef Ptak from Stal-Hurt S.C. company for providing sheets of Hardox steels.

**Conflicts of Interest:** The authors declare no conflict of interest.

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
