# Peer review of "Analysis of the Properties of Hardox Extreme Steel and Possibilities of Its Applications in Machinery"

_metals, doi:10.3390/met11010162_

Round 1

Reviewer 1 Report

REVIEW COMMENTS

Manuscript ID: metals-1078241

Analysis of the properties of Hardox Extreme steel and possibilities of its applications in machinery

A very interesting work. In addition, it is presented neatly and clearly, which makes it very pleasant to read.

The manuscript analyzes the chemical composition, mechanical properties and impact properties of Hardox Extrem steel. The experimental methodology used is described in detail and the results are presented and discussed clearly. However, some minor errors have been observed and some aspects could perhaps be better explained. For this reason, I would appreciate you consider the following suggestions:

  1. Line 53. There are other steelmakers that manufacture high strength abrasion resistant steels that can be used for weight reduction in heavy load carrying vehicles vehicles and machinery. An example could be Dillinger (https://www.dillinger.de/d/en/products/proprietary-steels/dillidur/). Likewise, the use of boron steels is common to reduce the weight of parts obtained by hot stamping of light vehicles, as is the case of Usibor steel manufactured by ArcelorMittal. (https://automotive.arcelormittal.com/products/flat/PHS/usibor_ductibor). The authors should do a more extensive analysis of high-strength steels from different steelmakers so that the paper does not look like an advertisement for SSAB.
  2. Line 109: “Taking into consideration the possibility of using welding techniques to connect the analysed steel”. The authors use the word "connect" to refer to welding joints throughout the paper. In my opinion, it would be more correct to use "join" instead of "connect". Please correct it throughout all the paper.
  3. Line 117: “…in the delivery state (after quenching) and in the normalizing annealed state”. In my opinion, it would be more correct to use "normalized state" instead of "normalizing annealed state". See Volume 4 of ASM Metals Handbook where it is explained that normalizing and annealing are different treatments.
  4. Line 127: What have been the criteria for selecting the austenitizing temperature (800 ºC)? Have the authors used any formula to calculate the Ac3 temperature, such as the one provided by J Traska, "Calculation of critical temperatures by empirical formulae", Arch Metall. Mater., Vol. 61, (2016)Nª 2B, pp. 981-986? Or have they carried out a trial and error test to determine it? Please explain the criteria considered to select the austenitizing temperature.
  5. Line 128: What is the size of the plate fragments that have been heat treated and what is the distance from the edges where the samples have been obtained to perform the different tests? I think it is interesting that the reader has this data to assess that the samples have been obtained from areas where the cooling rate is homogeneous.
  6. Line 140: Unlike other tests, the number of hardness measurements carried out on each type of sample is not indicated. However, the results indicate the standard deviation of the hardness measurement. Therefore, several measurements were made. Please indicate the number of hardness measurements performed on each type of sample.
  7. Line 144: Better to call "tensile tests" instead of "strength tests". As the authors explain, in these tests they obtain both strength properties, as well as ductility or stiffness of the steel.
  8. Line 145: I understand that 5mm diameter cylindrical tensile samples will be used. It is right? Does the axis of the samples coincide with the center of the plate? I suppose it is, but a clearer explanation of the geometry and location of the samples would be appreciated.
  9. Line 173: The nickel content of the analyzed steel is remarkable. It would be convenient for the authors to include a brief explanation of the possible reason of the addition of 2% Ni instead of, for example, a higher chromium content. I think the following comments collected in Volume 1 of the ASM Metals Handbook may be interesting for the authors. “Because nickel does not form any carbide compounds in steel, it remains in solution in the ferrite, thus strengthening and toughening the ferrite phase.” and “Chromium can be used as a hardening element, and is frequently used with a toughening element such as nickel to produce superior mechanical properties.”
  10. Line 204: “Destruction mechanisms”? I assume the authors are referring to failure mechanisms or failure modes. Correct it, please.
  11. Line 212: “Moreover, in both directions of rolling direction, one can observe bright bands, which indicates a slightly increased carbon content in these areas.” The presence of bright bands after etching could indicate differences in the carbon content of these areas. However, it is not clear to me that the bright areas correspond to those with a higher carbon content. They could also be due to the segregation of other alloying elements (eg Cr). Please provide a bibliographic reference that explains the observed phenomenon.
  12. Line 219: “Martensitic areas follow the heat-plastic transformation and are connected with martensite structure bands observed in the delivery state”. Sorry, I don't understand this sentence. What do the authors mean by “heat-plastic transformation”?. What are the “structure bands observed in the delivery state”?. The bright bands? Explain it more clearly, please.
  13. Line 253: What are the "two main strengthening mechanisms"? I think the authors do not indicate them or I do not understand the explanation.
  14. Line 254: What do the authors mean by “heat-plastic treatment”?. Thermomechanical rolling? Explain it more clearly, please.
  15. Line 261: Is it correct to indicate the elongation as A5? Or should A25 or A5d be indicated? Check it out, please.
  16. Line 283: “…impact strength changes in Hardox Extreme steel in the delivery state does not have a leap nature”. What do the authors mean with this sentence? That the impact behavior of steel is isotropic? Explain it more clearly, please.
  17. Lines 286-303: As can be seen in Figure 7, the value of the standard deviation is large for all tests. If this standard deviation is taken into account, instead of the mean value, it could be said that the results in the longitudinal and transverse directions are in the same range. Therefore, I do not know if the detailed description of the differences observed in the longitudinal and transverse direction is significant.
  18. Line 316: If I am not mistaken, the percentage of ductile zones in the fracture of normalized samples is not indicated. Please indicate it in order to compare with the samples in the delivery state.
  19. Lines 365-383: Could the brittle facets be related to the quenched martensite (without tempering) and the ductile fracture zones to the sorbite observed in the microstructure of the samples?
  20. Line 565-567: In my opinion, in references [8], [9] and [10], it would be more appropriate to indicate the link in English rather than Polish for a better understanding of the readers. For example: https://www.ssab.com/products/brands/hardox/customer-case-ecoupgraded-articulated-hauler

Author Response

Dear Reviewer,

We would like to thank you for your review. We are glad that you found the article interesting. It is very ennobling for us, especially since we have been testing boron abrasion-resistant steels for many years. Below are the responses to the review. All changes to the manuscript are emphasized.

With regards

On behalf of the other authors Beata Białobrzeska

Reviewer 2 Report

The paper presents the results of Hardox Extreme steel tests in the as-delivered state from a steel mill (after quenching and tempering), and also in the normalizing annealed state. The background of the steel has been described extensively in the introduction part, which give a good introduction to the steel. However, the scientific significance of the work has not been well addressed. The language is well and the results are comprehensive. It can be published in the Metals journal.

Before the paper can be published, some revisions should be made. Mainly in the Introduction part.

1)  Is there any other study which has reported the microstructure and the mechanical properties of the steel? what is the difference between this study and other work on the similar steels?

2)  What is the basic composition and microstructure of the steel?

3)  What is the kinetics of transformations?  e.g., the CCT curves or martensite starting temperature.

4)  Some experimental procedures, such as the composition analysis, which appeared in the Results section, should be put into the  Materials and methods section.

Author Response

Dear Reviewer,

We would like to thank you for reviewing our article and for believing it could be published in Metals. Below are the answers to your questions.

With regards

On behalf of the other authors Beata Białobrzeska
